# The Analysis of Present and Future Use of Non-Conventional Water Resources in Heilongjiang Province, China

Hongcong Guo [1] , Yingna Sun [1,*], Tienan Li [2], Yun Teng [2], He Dong [2], Hui Li [2] and Gengwei Liu [1]

[1] School of Hydraulic and Electric Power, Heilongjiang University, No. 74, Xuefu Street, Harbin 150080, China; 17839333953@163.com (H.G.); hss_liugengwei@126.com (G.L.)

[2] Heilongjiang Province Hydraulic Research Institute, No. 78, Yanxing Street, Harbin 150080, China; litienan0019@163.com (T.L.); hljskyty@126.com (Y.T.); albert121224@163.com (H.D.); hljsky_lh@163.com (H.L.)

* Correspondence: 2006090@hlju.edu.cn

**Abstract:** Analyzing the development trend of non-conventional water resources and identifying the main influencing factors is the initial step toward rapidly increasing the utilization and allocation of these resources in a rational and scientific manner. This will help relieve pressure on water resources and improve the ecological environment. This study introduces the concept of comparison testing and employs advanced Dematel and Random Forest models to identify two sets of optimal indicators from a pool of nine. Based on the two best indicator sets, three prediction models—BP neural network, Particle Swarm Optimization-optimized BP neural network, and Genetic neural network—were used to forecast the future potential of non-conventional water resource use in Heilongjiang Province. The findings reveal that economic indicators are the most significant factors influencing Heilongjiang Province's utilization of non-conventional water resources. The findings of this study help us understand the extent of development in utilizing non-conventional water resources.

**Keywords:** non-conventional water resources; key indicator screening; potential prediction; neural network; optimization algorithm

## 1. Introduction

The evolution of human society and economy depends on water resources [1]. There is currently a global scarcity of water resources and an insufficient sustainable water supply due to the rapidly increasing population, the fast-developing socioeconomic sector, and the impacts of climate change [2–5]. China is among the nations in the world with a mild water deficit, despite having an adequate overall amount of water resources because of its massive population and exceptionally low per capita possession of water resources. China will continue to struggle with an imbalance between the availability and demand of water resources in the coming decades [6]. China has very little freshwater resources—only 10% of the world's total—according to the United Nations Environment Program's Global Environment Outlook (GEO) report. One of the primary barriers to China's overall growth is the scarcity of freshwater, which is essential for socioeconomic advancement [7–9]. Non-conventional water resources are increasingly becoming a supplementary water source for urban areas, thanks to advancements in science and technology. This is a significant factor in reducing urban water scarcity and promoting water recycling. Therefore, the keys to solving the issues lie in increasing the pace at which non-conventional water resources are utilized and allocating these resources in a fair, rational, equitable, and scientific manner [10].

Treated wastewater, treated agricultural drainage, desalinated water, brackish groundwater, and harvesting of rainfall are examples of non-conventional water supplies. In industrial, municipal, and agricultural irrigation, recycled water is a competitive substitute for traditional water supplies. It is essential for reducing water shortages and mitigating urban water pollution. It also offers a great deal of room for development. In the modern

era of water resource management, utilizing non-conventional water sources is crucial for implementing the concept of "prioritizing water conservation [11,12]". China now primarily utilizes non-conventional water sources for agricultural, forestry, groundwater recharge, urban non-potable water systems, industrial settings, and landscaping. But the treatment approaches are relatively outdated, and the dissemination strategies are not perfect [13,14]. China should actively develop and utilize non-conventional water resources by integrating them with the country's specific circumstances. It is essential to study the treatment and utilization methods of non-conventional water resources in countries such as the United States, Japan, Australia, Iran, and Spain [15]. China's Heilongjiang Province is a significant agricultural region. However, in recent years, there has been a drastic reduction in rainfall, leading to the majority of the province's river water bodies being classified as IV and V, indicating generally poor river water quality. In brief, the province of Heilongjiang is currently facing several challenges, such as poor water quality, unequal distribution of resources, and reduced water availability. Thus, it is imperative that non-traditional water resources be included in Heilongjiang Province's total water resource allocation.

Previous studies have indicated that some researchers have attempted to identify environmental factors and water quality as influential factors. They utilized Spearman correlation and redundancy analysis to identify the main influencing factors [16,17]; Wu considered water resource endowment conditions and water resource utilization as driving factors, while water conveyance engineering, economy, and ecology were viewed as constraints. Additionally, ecological health was identified as a risk factor. Wu created an evaluation based on the rough set theory to determine weights, providing a robust evaluation index system for non-conventional water resources [9]. The allocation of non-conventional water resources utilization can be categorized into several stages: production, distribution, end-use, and wastewater treatment processes [18,19]. These processes are widely influenced by regional factors such as climate change and topography, social factors including urbanization and population growth, behavioral factors like water use patterns and operational efficiencies, and managerial factors. Therefore, the factors affecting the use of non-conventional water resources are typically categorized into social, economic, and environmental factors as well [20,21]. The allocation of water resources for Chinese cities often has a direct relationship with the regional urban development plan and water resources. Therefore, it is crucial to focus on water resources, water consumption levels, urban development plans, economic indices, and other indicators when determining the key factors that influence the utilization of non-conventional water resources [22,23]. This is because economic variables are often closely linked to the extent of utilization of non-conventional water resources. Recent studies indicate that economic factors are key influencers in the utilization of non-conventional water resources. The use of non-conventional water sources in various sectors contributes to economic and social progress, often resulting in undisclosed economic benefits at different levels. Additionally, regions with rapid economic development typically have more resources available to support their allocation [24,25]. Economic factors will be the most significant variables influencing the potential utilization of non-conventional water resources, based on the extrapolation of findings from earlier studies [26]. Before implementing non-conventional water use, it is essential to examine the key indicators of its impact [27]. Comprehensive consideration should be given to the cost of water extraction and distribution, regional development priorities, and the economic viability of each approach [28]. In order to lay the groundwork for the future allocation of non-conventional water resources in industry, agriculture, domestic use, and ecological recharge, this paper aims to identify the key factors that influence the utilization of non-conventional water resources. Additionally, it predicts the potential use of non-conventional water resources based on highly correlated indicators [29].

This study introduces the theory of comparison experiments and establishes a more comprehensive set of impact index systems for measuring the degree of non-conventional water resource usage, based on previous research and real investigations [30,31]. The control group consisted of the Random Forest–Back Propagation neural network (Random Forest–

BP neural network) and the Improved Decision-making Trial and Evaluation Laboratory–Back Propagation neural network (Improved Dematel–BP neural network), while the study also utilized the Improved Decision-making Trial and Evaluation Laboratory–Particle Swarm Optimization–Back Propagation (Improved Dematel–PSO–BP), Improved Decision-making Trial and Evaluation Laboratory–Genetic Algorithm–Back Propagation (Improved Dematel–GA–BP), Random Forest–Particle Swarm Optimization–Back Propagation (Random Forest–PSO–BP), and Random Forest–Genetic Algorithm–Back Propagation (Random Forest–GA–BP) neural networks. In this study, the BP was optimized by the GA and PSO as a way to increase the global optimization capability of the data simulation and improve the fitting accuracy. The experimental group's decision making regarding the significant impact of non-conventional water resources utilization in Heilongjiang Province is based on the error of the Rokuzu experiment's fitting process, the error of the prediction results, and the actual value of the prediction year (2023) for the two groups of data for comparison. The objective is to achieve the most economical result [32–34]. The study's flow is depicted below (Figure 1).

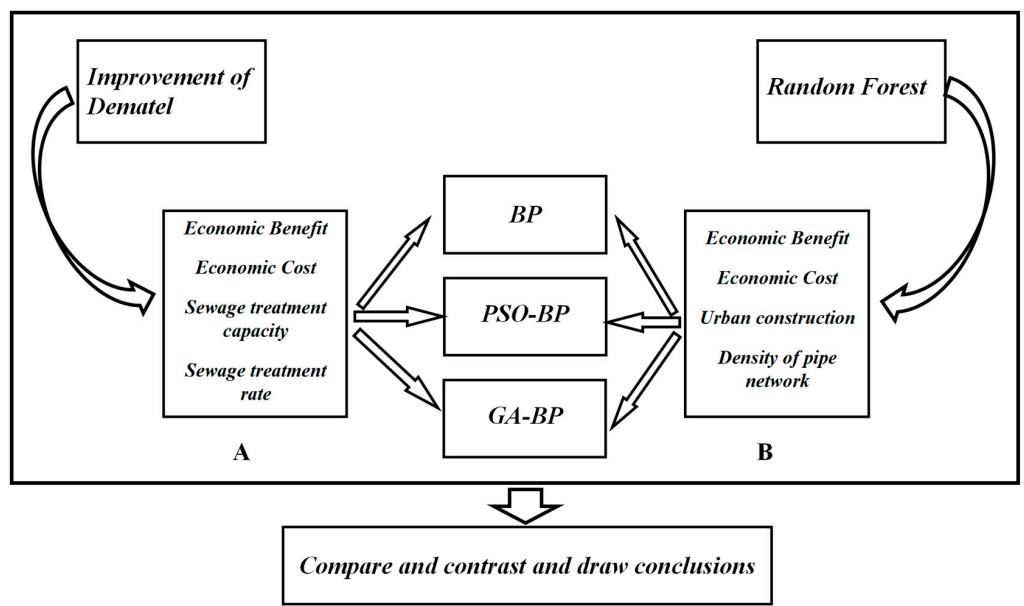

**Figure 1.** Flowchart for the research.

The use of non-conventional water sources in Heilongjiang Province contributes to the sustainable development of the region. Before allocating non-conventional water resources, it is important to understand the key factors that affect the utilization of non-conventional water resources and to forecast their potential utilization in order to formulate a reasonable allocation plan in advance. Therefore, this study compares various models to identify the most significant factors influencing the utilization of non-conventional water resources in Heilongjiang Province. It also predicts the potential for non-conventional water resource utilization and compares the predicted results with the actual values. This is performed to select the indicator system and prediction model that best align with the requirements of Heilongjiang Province.

## 2. Materials and Methods

### 2.1. Overview of Water Resources in Heilongjiang Province

Heilongjiang Province in Northeastern China covers a total area of $47.3 \times 10^4$ km$^2$ and is situated between 121°11′–135°05′ E and 43°26′–53°33′ N. The Songhua River, Heilongjiang River, Ussuri River, and Suifen River are the four main rivers in the region. During the growing season, which spans from May to September, precipitation can account for 80% to 90% of the annual total. The average annual precipitation usually ranges from 400 to

650 mm. The central mountainous regions receive the highest amount of rainfall, followed by the east. The western and northern regions experience the least amount of precipitation. During the summer, when the southeast monsoon is active, there is a substantial amount of precipitation, contributing to approximately 65% of the annual rainfall. In the winter, the dry and brisk northwest winds prevail, resulting in minimal moisture and snowfall, which make up only 5% of the annual precipitation. In the spring and fall, the precipitation levels account for approximately 13% and 17% of the annual total, respectively. July was the wettest month, while January had the least rainfall. The overall depiction includes a chilly and dry spring, a hot and rainy summer, and a flood-prone fall with early frosts. The total water resources in Heilongjiang Province have been steadily increasing each year, as reported in the 2015–2020 Water Resources Bulletin. Although Heilongjiang Province has an overall abundance of water resources, water pollution has become a severe problem in recent years, and many businesses have very high water demands [35–37]. The map of the Heilongjiang water system is shown in Figure 2.

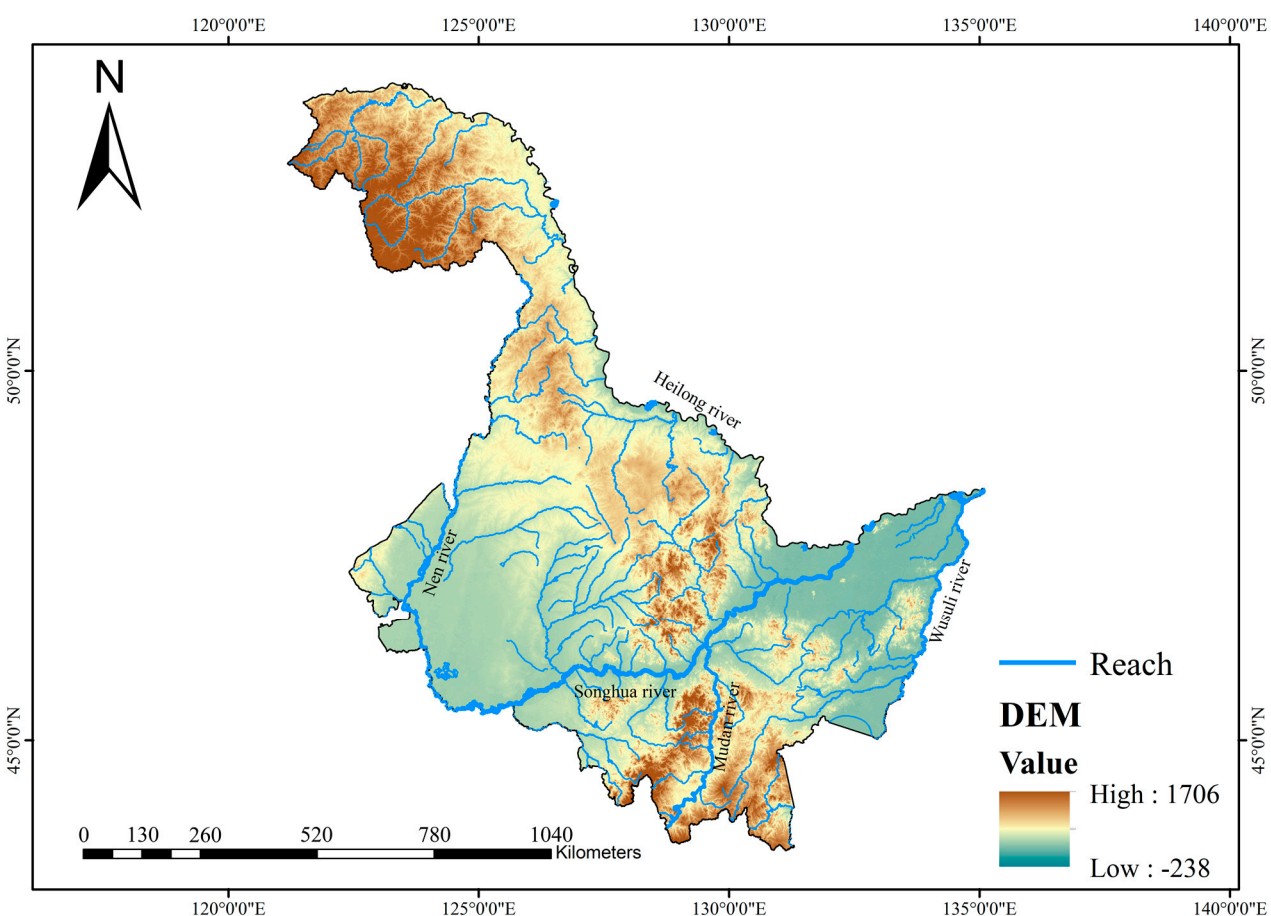

**Figure 2.** Map of the Heilongjiang Water System. The word "Reach" in the figure indicates a river; data from "Heilongjiang Province Water Resources Bulletin".

## 2.2. Selection of Indicators

In Heilongjiang Province, non-conventional water sources are mainly used for domestic water consumption, industrial manufacturing, agricultural irrigation, and ecological water restoration. Nine variables are selected, including economic advantages, economic costs, water demand, and the percentage of urban population, to consider the impacts of economic, social, and urban development. In the context of ecological recharge, the economic benefits are assessed using the "hedonic price method", which quantifies the ecological environment based on the value people are willing to pay for a high-quality environment. The economic benefits for industry, agriculture, and livelihoods include

both production value and water tariffs [38]. In this study, we quantify the economic benefits of non-conventional water use for ecological recharge by analyzing the statistical increase in housing prices and the population's consumption capacity in Heilongjiang Province, which can be attributed to environmental improvements. The cost of constructing a pipeline system, maintaining a pipeline network, and treating non-conventional water are all considered economic indicators. Table 1 presents the unique indicator system.

**Table 1.** Indicators of the effects of non-conventional water use in the province of Heilongjiang.

| Norm | Formula | | Formula No. | Clarification |
|---|---|---|---|---|
| Economic benefit ($X_1$) | Agriculture: $$E_f = \sum_{i=1}^{3} y_i \times (p_i + \xi_i w_i)$$ | $X_1 = E_f + E_s + E_l + E_e$ | (1) | "i" represents the type of agriculture, "y" represents production, "p" represents the unit price of the product; "w" represents the price of water; "ξ" represents the amount of water used per unit of product. |
| | Industry: $$E_s = \sum_{j=1}^{7} (y_j \times p_j + m_j \times w_j)$$ | | (2) | "j" represents the industry type, while "m" represents the amount of water used per unit of industrial output. |
| | Resident life: $$E_l = \sum_{l=1}^{2} n_l \times q_l \times w_l$$ | | (3) | "l" represents the population type (urban and rural); "n" represents the number of people; "q" represents the water quota. |
| | Ecological: $$E_e = \sum_{k=1}^{3} [n_k \times C_k \times (1 + \partial_k)] + \sum_{o=1}^{3} [m_o \times C_o \times (1 + \partial_o)]$$ | | (4) | "k" represents the level of consumption; "o" represents the level of house prices; "$C_k$" represents the capacity to consume; "$C_o$" represents the level of house prices; "m" represents the size of housing; "∂" represents the growth rate |
| Economic cost ($X_2$) | $$X_2 = l \times (p + b) + \sum_{u=1}^{3} U_u \times g_u$$ | | (5) | "l" represents the length of the pipe; "p" represents the unit price; "b" represents the unit maintenance cost; "u" represents the non-conventional water type; "U" represents the treatment capacity; "g" represents the unit treatment cost. |
| Water demand potential ($X_3$) | $$X_3 = y_e + \sum_{i=1}^{3} s_i \times \delta_i + \sum_{j=1}^{7} g_j \times \delta_j + \sum_{l=1}^{2} n_l \times \delta_l$$ | | (6) | "$y_e$" represents ecological water use; "s" represents the area of agricultural land; "g" represents industrial output; "δ" represents the water quotas. |
| Water supply capacity ($X_4$) | $$X_4 = \sum_{k=1}^{2} w_k$$ | | (7) | "k" represents the type of water supply, unless conventional water, the water supply channels in Heilongjiang Province are surface water and groundwater. |
| Urban construction ($X_5$) | $$X_5 = \frac{S_c}{S_t}$$ | | (8) | "$S_c$" represents the built-up area of the urban area; "$S_t$" represents the total area. |
| Percentage of population in urban areas ($X_6$) | $$X_6 = \frac{P_c}{P_t}$$ | | (9) | "$P_c$" represents the urban population; "$P_t$" represents the total population. |
| Sewage treatment capacity ($X_7$) | $$X_7 = \sum_{f=1}^{n} UW_f$$ | | (10) | "n" represents the number of wastewater treatment plants; "UW" represents the volume of wastewater treated. |
| Sewage treatment rate ($X_8$) | $$X_8 = \frac{uw}{se}$$ | | (11) | "$S_e$" represents the volume of sewage discharged. |
| Density of pipe network ($X_9$) | $$X_9 = \frac{s_t}{S_c}$$ | | (12) | "$S_t$" represents the area covered by the pipe network; "$S_c$" represents the total area of the region. |

For Equation (3), the residents of Heilongjiang are divided into urban and rural populations, and the water use benefit of the total population is calculated according to the water use quota of the two different types of residents [39]; for Equation (4), the "hedonic price method" is invoked to calculate the promotion effect of ecological water replenishment on peripheral consumption [40]; for Equation (5), the main types of unconventional water resources in Heilongjiang include reclaimed water, rainwater, and mine water, and the three types of unconventional water sources have different treatment costs. The cost of unconventional water use is the sum of transport costs, management costs, and treatment costs [41]; for Equation (6), the sum of water consumption for four types of industries: agriculture, industry, life, and ecology; Equation (7) indicates the total amount of surface water and groundwater that can be allocated to a certain industry [42]; Equation (8) indicates the level of urbanization in each region, i.e., the ratio of the area of built-up area to the total area; Equation (9) indicates the level of urbanization in each region, i.e., the proportion of urban population to the total area and development level of each region, i.e., the proportion of the urban population to the total population; Equation (10) represents the total water purification capacity of all sewage treatment plants in Heilongjiang Province; Equation (11) represents the level of sewage treatment in Heilongjiang Province, i.e., the proportion of the treatment volume to the total volume of sewage discharged [43]; Equation (12) represents the level of construction of the official website for water transport in Heilongjiang Province.

### 2.3. Selection of Important Indicators

Identifying the key impact indicators of non-conventional water usage potential is crucial for water-scarce regions aiming to substantially enhance the utilization of non-conventional water resources. This process assists these regions in understanding how to elevate the level of non-conventional water resource utilization and is a prerequisite for predicting the potential and rational allocation of non-conventional water resources.

Decision-making Trial and Evaluation Laboratory (Dematel) [44] is a model used to determine the importance of each influencing factor. In this study, Kenda II was utilized to improve the Dematel analysis [45] and eliminate the influence of subjective behavior.

The Random Forest model assesses feature importance by comparing the magnitude of contribution between different features [46]. The sample size for this study is 14, and the survey collects data on variables in Heilongjiang Province from 2008 to 2021.

Figure 3 represents the Dematel model and Figure 3 represents the Random Forest model. The workflow for the improvement of the Dematel model and the Random Forest model is shown in Figures 3 and 4.

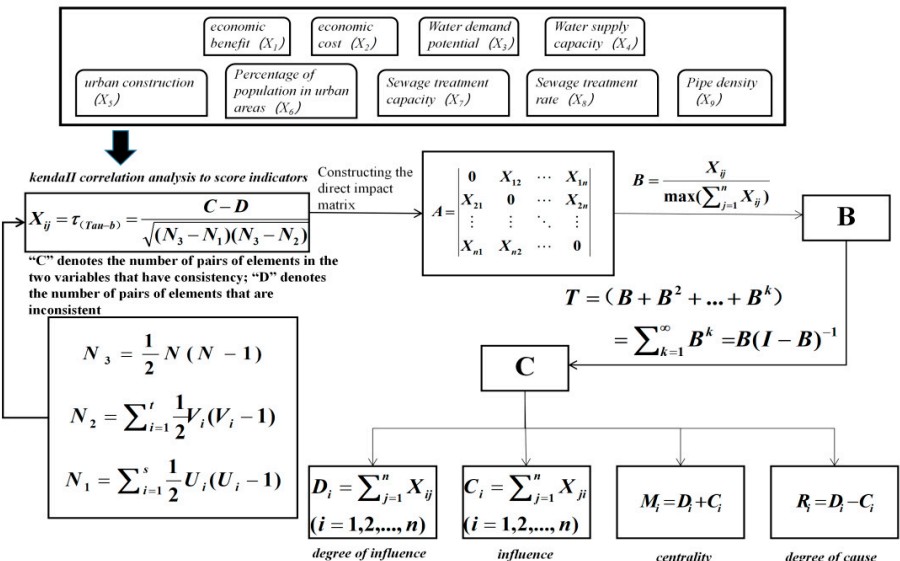

**Figure 3.** Improvement of the Dematel model for screening variable flow [47,48].

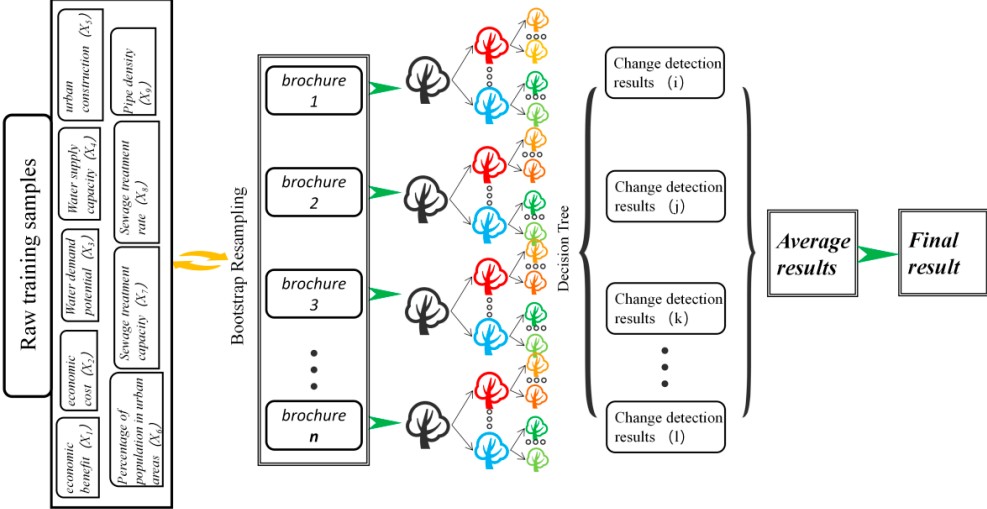

**Figure 4.** Random Forest model for variable screening process.

*2.4. Forecasting the Potential for Non-Conventional Water Use*

Prediction of the potential for non-conventional water utilization is essential for allocating non-conventional water resources. This study combines the key indicators influencing the potential utilization of non-conventional water resources that were identified previously. It then uses BP neural network, PSO-BP, and GA-BP to simulate and predict the potential of non-conventional water resource utilization. This approach allows for the assessment of the level of development of non-conventional water resource utilization in Heilongjiang Province and provides a basis for future enhancement of non-conventional water resource utilization in the region.

2.4.1. BP Neural Network

A widely used artificial neural network architecture known as the Back Propagation Neural Network (BPNN) provides exceptional learning and adaptation capabilities [49]. The study's progression is depicted below (Figure 5):

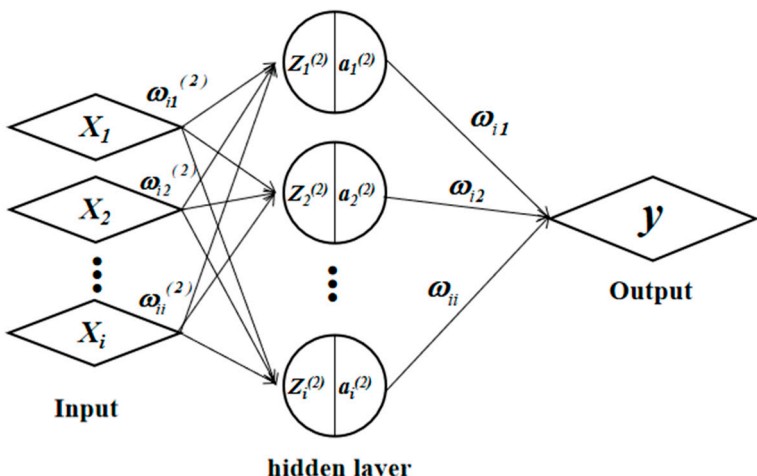

**Figure 5.** BP neural network prediction process. In the figure, "$\omega$" denotes the weight parameter; "Z" denotes the input of the implicit layer; and "$\alpha$" denotes the output of the implicit layer.

The BPNN is optimized using both the Particle Swarm Algorithm and Genetic Algorithm. The parameters are iteratively optimized to achieve the most accurate predictions. This is because the BPNN is unable to determine whether the network structure is appropriate, if the parameter settings are reasonable, if the training is sufficient, or if there are any other issues in the working process.

2.4.2. Prediction of Non-Conventional Water Use Potential Based on Optimization Algorithm BPNN Models

In this study, Particle Swarm Optimization (PSO) [50] and the Genetic Algorithm (GA) [51] are applied to optimize the BPNN, respectively. The two optimal sets of indicators screened in Section 2.2 are selected as input variables to establish a BPNN. The PSO and GA algorithms are used to iterate the neural network in order to obtain the optimal results.

The progression of the study is depicted below (Figure 6):

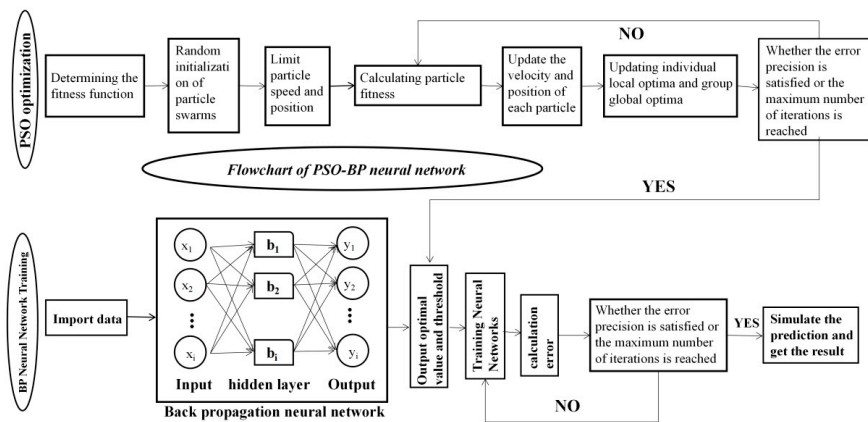

I. PSO-BP prediction process.
(PSO's optimization process for BPNN is represented at the top of the figure)

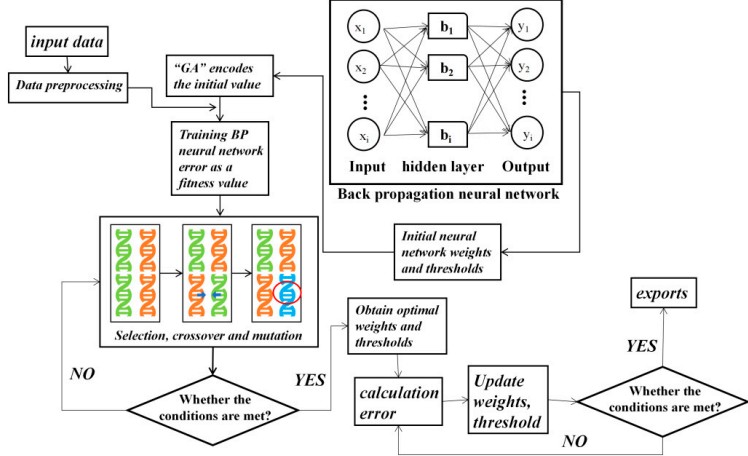

II. GA-BP prediction process.
(In the figure, arrows and circles indicate gene exchange and mutation, respectively)

**Figure 6.** Optimization algorithm-based BP neural network prediction process.

### 2.5. Modeling of Non-Conventional Water Allocation in Heilongjiang

"Sansheng" refers to production, living, and environment, and this sector often utilizes non-conventional water sources. The primary objective of this study is to maximize economic benefits. The main constraints for optimizing the allocation of non-conventional water resources in Heilongjiang Province during the planning year of 2025 include water balance constraints, supply and demand constraints of water use units, and non-negativity constraints. These constraints were determined based on the screening of key indicators, which revealed that economic factors were the most important influencing factors of non-conventional water resource utilization.

$$f_{1\max} = \sum_{i=1}^{13} (\alpha_i - \beta_i) \cdot X_i \tag{13}$$

where "$\alpha_i$" denotes the production efficiency coefficient for each region; "$\beta_i$" denotes the cost coefficient for each region; and "$X_i$" denotes the amount of non-conventional water resources utilized.

$$ST \begin{cases} \sum\limits_{i=1}^{13} X_i \leq (1-\lambda)Q \\ Q_{i\min} \leq X_i \leq Q_{i\max} \\ Q_{j\min} \leq \sum\limits_{j=1}^{4} X_j \leq Q_{j\max} \\ Q_{k\min} \leq \sum\limits_{k=1}^{13} X_k \leq Q_{k\max} \\ X_i \geq 0 \end{cases} \qquad (14)$$

In the formula, "ST" denotes constraints; "$\lambda$" indicates the water loss coefficient (%) of the water transmission and distribution process; "Q" indicates the maximum available water supply; "j" denotes the water use sector; and "k" denotes the water use unit.

## 3. Analysis of Results

### 3.1. Screening Results

The data for each index were calculated by considering the degree of non-conventional water resource use in Heilongjiang Province from 2008 to 2021. Then, Kenda II was used to determine the scoring matrix A.

To screen nine indicators, the Improved Dematel model and the Random Forest model were chosen. The Improved Dematel model and the Random Forest model were considered central in assessing the impact of indicators related to the potential of utilizing non-conventional water resources in Heilongjiang Province. I simulated the data by using the MATLAB2022 software, and the level of influence of the resulting indicators was displayed in Figure 7.

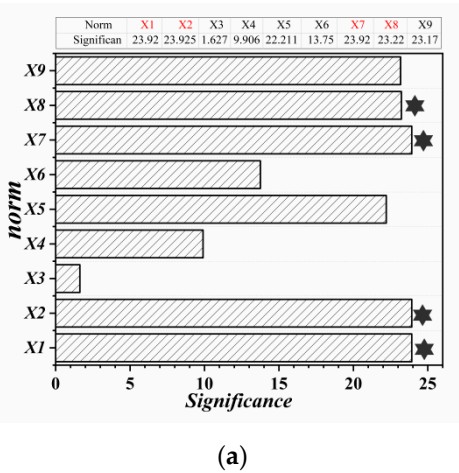

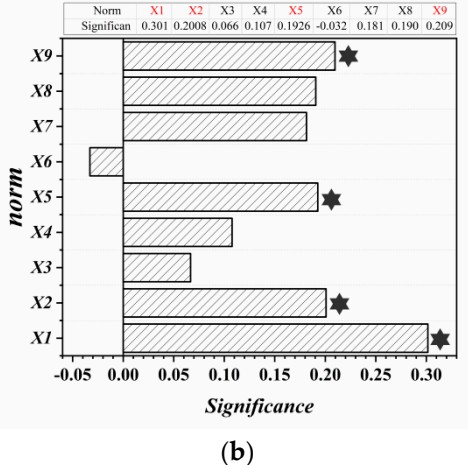

(**a**)          (**b**)

**Figure 7.** Vitality of indicators. The red font and "*" in the graphic represent the main indicators that have been screened out; (**a**) denotes the centrality of each indicator, which has been enhanced by the Dematel model, and (**b**) indicates the importance of each indicator obtained by the Random Forest model.

By enhancing the Dematel and Random Forest models, two sets of critical indicators were tested, as shown by the findings, namely $|A| = [X_1, X_2, X_7, X_8]$ and $|B| = [X_1, X_2, X_5, X_9]$. The PSO-BP and GA-BP models were then used to fit the BP neural network and forecast the likelihood of utilizing non-conventional water resources in Heilongjiang Province, respectively.

### 3.2. Predicted Results

Two sets of ideal variables were chosen ($|A|, |B|$) based on the findings in Section 3.1. MATLAB2022 software was then used to simulate data for prediction. As the training set, the data for each indicator in Heilongjiang Province from 2008 to 2017 was input, and as

the test set, the data for each indicator from 2018 to 2021 was input. The fitting process and prediction results are shown in the Figures 8 and 9.

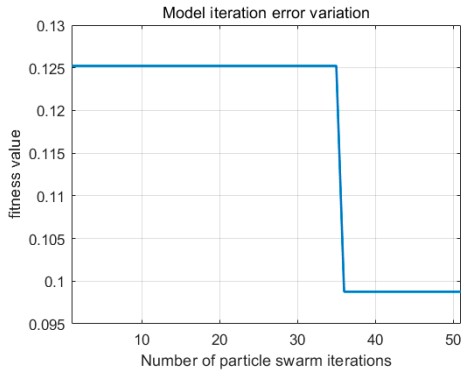

Improved Dematel-PSO-BP model iteration error

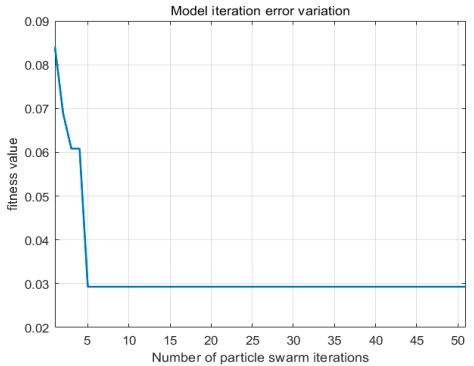

Random Forest-PSO-BP model iteration error

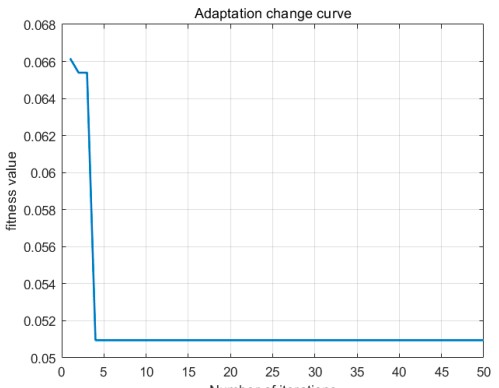

Improvement of the Dematel-GA-BP adaptation change curve

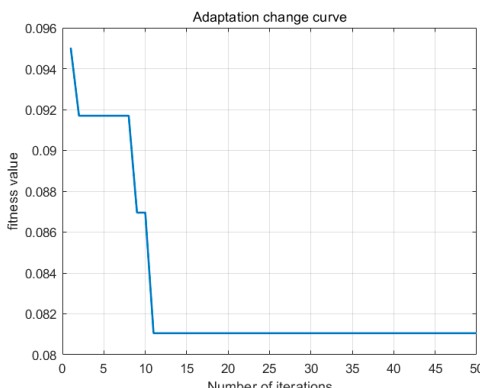

Random Forest-GA-BP adaptation change curve

**Figure 8.** Fitting process.

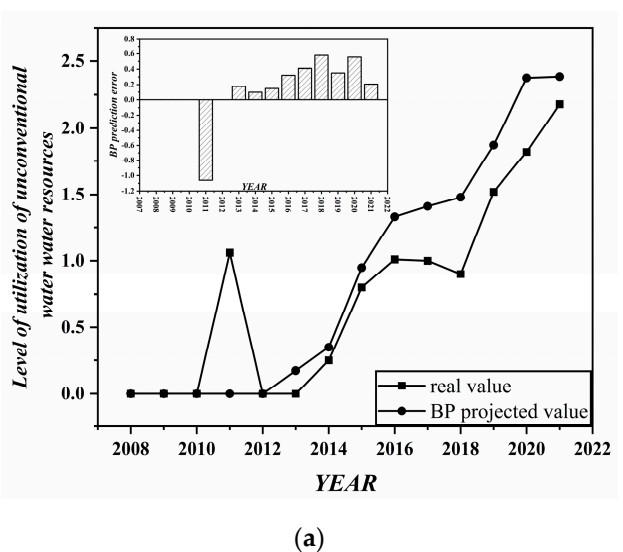

(**a**)

(**b**)

**Figure 9.** *Cont.*

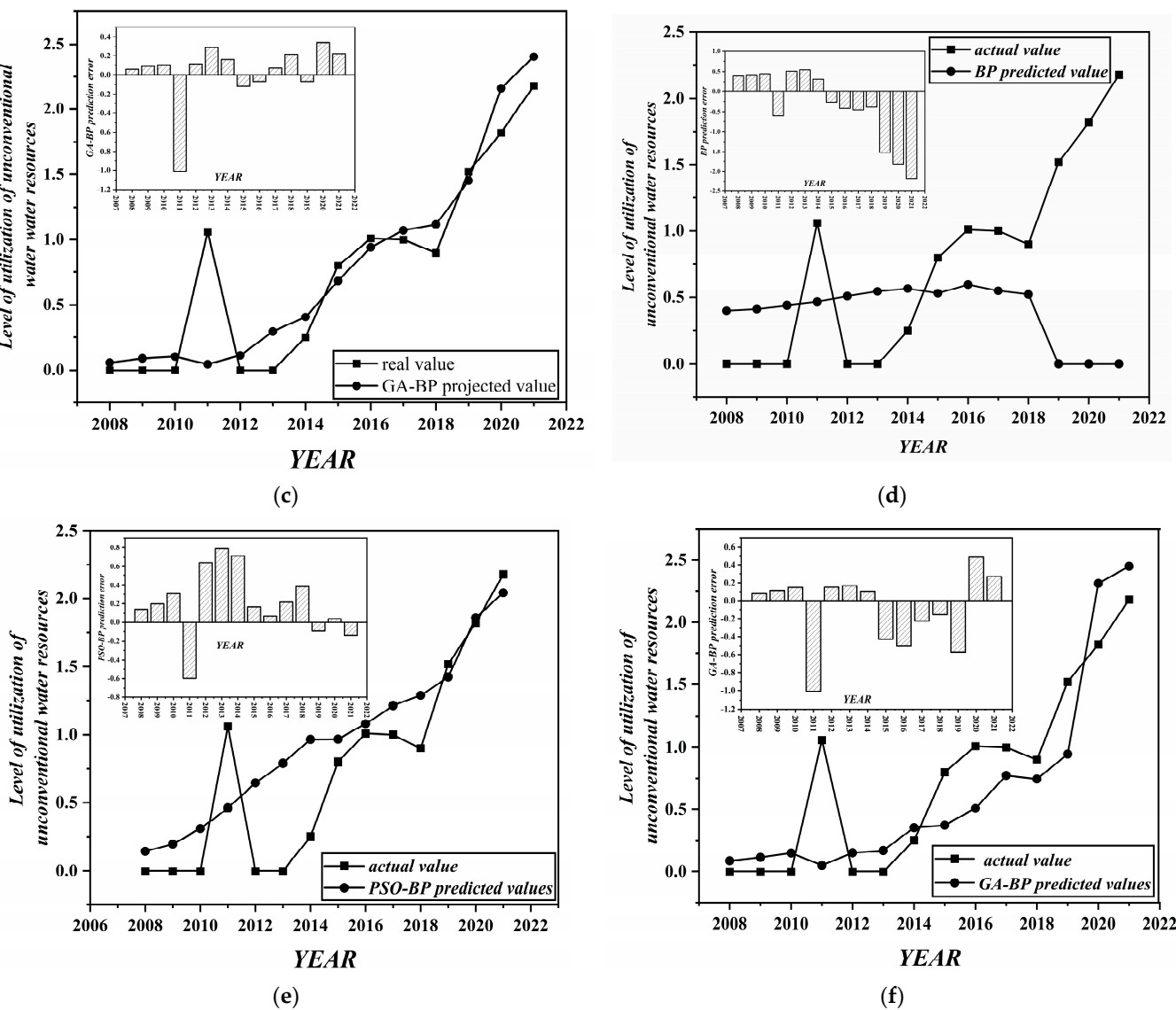

**Figure 9.** Prediction error. (**a**–**c**) are the prediction errors for the three models of BP, PSO-BP, and GA-BP on the index set $|A|$, and (**d**–**f**) are the simulation curves and errors of the three models of BP, PSO-BP, and GA-BP on the index set $|B|$, respectively.

The three prediction models generate a total of six fitted curves based on the two sets of optimal variables. These curves are then used to analyze the data of the indicators for Heilongjiang Province in 2022 and determine the potential for utilizing non-conventional water resources. I then compare the projected potential for unconventional water use with the actual utilisation of unconventional water resources in the validation year (2022), as depicted in Figure 10.

As depicted in the figure, the key indicators identified by the Improved Dematel model show that the non-conventional water resource utilization in Heilongjiang Province in 2022, predicted using the GA-BP neural network, is the most suitable for the actual conditions. Therefore, the Improved Dematel-GA-BP model is the most aligned with the prediction model for the potential utilization of non-conventional water resources in Heilongjiang Province.

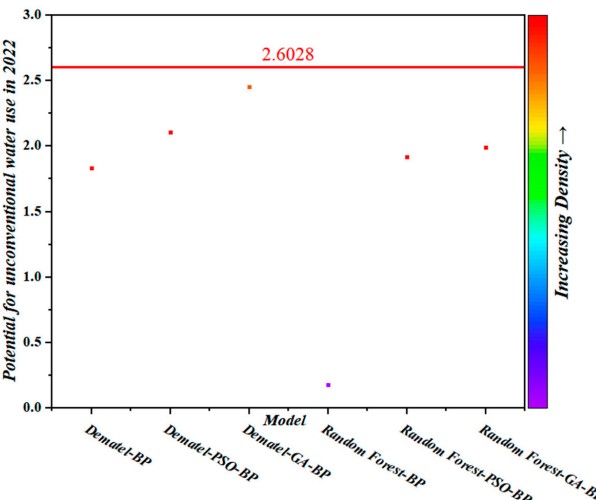

**Figure 10.** Comparison of model predictions with actual values. In the figure, "2.6028" is the actual utilization of non-conventional water resources in Heilongjiang Province in 2022.

### 3.3. Configuration Results

The Dematel model establishes an optimal allocation model with the objective of maximizing economic benefits. The GA-BP model predicts the non-conventional water resources of Heilongjiang Province in the planning year 2025. The optimal allocation scenario (OS scenario) is calculated. The above combinations represent the best screening and predictive methods for identifying factors influencing non-conventional water resources in Heilongjiang Province. As per the OS scenario, Heilongjiang Province will have $3.3 \times 10^8$ m$^3$ of dispatchable non-conventional water resources in 2025. Jiamusi has been allocated the highest amount of non-conventional water resources overall among water use units, accounting for up to 22% of the total; Qiqihar has been allocated the highest amount of industrial non-conventional water among water use sectors, representing up to 2.05%; Jiamusi has been allocated the highest amount of agricultural non-conventional water, accounting for up to 21.02%; and Harbin has the highest amount of domestic and ecological non-conventional water, with 1.34% and 0.38%, respectively. Figure 11 illustrates the non-conventional water allocation strategy for each unit and each water sector.

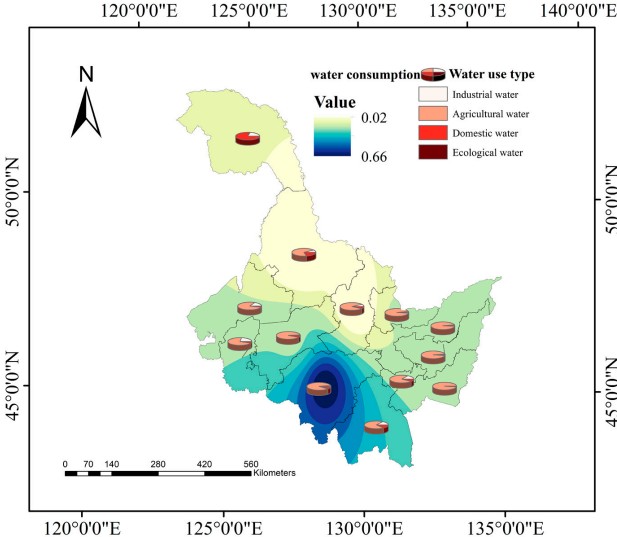

**Figure 11.** Non-Conventional Water Resources Allocation Program for Heilongjiang Province in OS programs.

## 4. Discussion

(1) The selection of indicator layers is diverse, often encompassing the economic perspective, city size, water resource availability, water usage levels, and other relevant factors. The evaluation of the use and allocation of non-conventional water resources has not yet established a consistent standard [52]. Economic indicators, such as the financial benefits and costs, consistently emerge as the primary factors influencing the use of non-conventional water resources. This conclusion is based on the screening results of nine indicators using the Improved Dematel and Random Forest model. This finding is consistent with the study's initial hypothesis. The construction of sewage and rainwater collection, treatment, and transportation facilities is closely linked to the utilization of non-conventional water resources [53]. Moreover, in a social and economic context, the level of profitability also directly impacts the willingness of citizens to utilize non-conventional water supplies [54]. The extent of urban infrastructure development is closely linked to the use of non-conventional water resources. Therefore, increasing investment in non-conventional water resources, improving urban technical facilities, and enhancing the management of non-conventional water resources can all significantly increase the utilization of non-conventional water resources in Heilongjiang Province [39]. In this study, nine indicators of economic and social factors were considered in the selection of indicators for determining the potential of non-conventional water resource utilization, and the results showed that the economic factors and the level of urban infrastructure construction are the most critical. This result is a guide for the direction of work to enhance the level of non-conventional water resource utilization in Heilongjiang Province. However, the problem of water allocation involves the role and reaction of three social factors, namely producers, consumers, and managers. In the next study, a control system for the role and reaction of producers, consumers, and managers should be established by combining environmental and sociological theories, exploring the promotional and inhibitory effects of economic factors on producers and consumers, as well as the effects of managers' coercive forces on producers and consumers, in order to optimize the efficient use of non-conventional water resources.

(2) The utilization of non-conventional water resources in Heilongjiang Province shows a generally increasing trend, as indicated by the projection results of the six sets model. The traditional BP neural network should be optimized because, based on the results of the BP neural network fitting process, the method exhibits a significant amount of error. This incorrect result is caused by the significant fluctuation in the output data of the dataset and the failure to optimize the weights and thresholds of the input data. These factors lead to deviations in the training process. Based on the evaluation of the six sets of prediction results, it is found that the Dematel–GA–BP model is the most suitable for the non-conventional water resource utilization in Heilongjiang Province, which can achieve the accurate prediction of the potential of non-conventional water resource utilization in Heilongjiang Province in the future, and also provide a good foundation for the non-conventional water resources allocation problem.

(3) To address water scarcity in Heilongjiang, it is crucial to assess the potential of using non-conventional water resources and understand the factors that influence their utilization [55]. One effective strategy for managing water scarcity is to integrate non-conventional water resources into the unified water resources supply system [56]. Non-conventional water resources can serve as a sustainable alternative and enhance water supply in vital industries. In this scenario, public policy should support the implementation of such practices [57]. Improving the utilization of non-conventional water resources in Heilongjiang Province can alleviate the pressure on the region's water resources. Additionally, using non-conventional water resources for replenishing rivers and lakes, as well as for public areas, can increase economic returns. Therefore, to enhance the utilization of non-conventional water resources in Heilongjiang Province, this initiative is crucial for achieving the establishment of a globally "environmentally friendly and resource-efficient society", which promotes the healthy and high-quality development of the world. It is

a significant action based on the shared values of society, the world, and humanity as a whole. It holds great significance for the establishment of a shared destiny for humanity and for the advancement of a better future for our collective homeland [58].

(4) In Heilongjiang Province, the proportion of agricultural water allocation in non-conventional water resources is significantly higher than that for industrial, domestic, and ecological water replenishment. This finding indicates the province's strategic position as a major agricultural producer. Consequently, the strategic direction of the region's growth should be considered when allocating water resources, rather than solely focusing on economic considerations. This aspect will also be the subject of the following work.

## 5. Conclusions

Utilizing non-conventional water resources is a significant solution to address the water shortage issue in Heilongjiang Province. In this study, the key indicator set was screened using Improved Dematel modeling $|A|$ (economic benefits ($X_1$), economic costs ($X_2$), sewage treatment capacity ($X_7$), and sewage treatment rate ($X_8$)), and the set of key indicators were screened by Random Forest modeling $|B|$ (economic benefits ($X_1$), economic costs ($X_2$), urbanization ($X_5$), and pipeline density ($X_9$)). Two sets of optimal indicators are used to predict, using a BP neural network, the future potential of using non-conventional water resources in Heilongjiang Province; PSO and GA algorithms are then applied, respectively, to optimize the BP neural network, and the results are then compared and analyzed in accordance with Rokuzu's prediction results. The research demonstrates that the models' prediction efficacy follows the order: Random Forest–GA–BP > Random Forest–PSO–BP > Dematel–PSO–BP > Random Forest–BP > Dematel–BP. According to the results of non-conventional water resource allocation in Heilongjiang Province, among the thirteen cities, except for Daxinganling, the non-conventional water resource allocation is mainly concentrated in agriculture, while Daxinganling, as a forestry conservation area, the proportion of agricultural non-conventional water resource allocation is basically zero. This allocation scheme is in line with the regional development model of Heilongjiang Province.

**Author Contributions:** H.G. contributed to conceptualization, methodology, software, formal analysis, investigation, writing—original draft, writing—review and editing, and visualization. Y.S. contributed to writing—review and editing and funding acquisition. T.L. contributed to writing—review and editing, resources, and funding acquisition. Y.T. contributed to writing—review and editing, resources, and funding acquisition. H.D. contributed to investigation, supervision, and resources. H.L. contributed to project administration and investigation. G.L. contributed to project administration, contributed to investigation. All authors have read and agreed to the published version of the manuscript.

**Funding:** National Key R&D Program of China (2022YFD1500402).

**Data Availability Statement:** All data and materials support the published claims and comply with field standards.

**Conflicts of Interest:** The authors declare no conflict of interest.

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
