# Peer review of "The Analysis of Present and Future Use of Non-Conventional Water Resources in Heilongjiang Province, China"

_sustainability, doi:10.3390/su16093727_

Round 1

Reviewer 1 Report (Previous Reviewer 1)

Comments and Suggestions for Authors

Your response seems too brief. Although you have annotated your modifications within the text, you still need to briefly describe your changes in the review comments for our review. The sections on methodology explanation and result validation still need improvement. Additionally, the formatting of your text is still quite poor, with multiple instances of inconsistent formatting. Further careful review and editing of the article are needed.

 Here are our major comments:

1)          In lines 95-98 of P2, the statement "The control group..." mentions various treatment methods without specifying how these methods differ from previous research methods, what problem they aim to address, and it doesn't clarify the full names of abbreviations such as DEMATEL, BP, GA, PSO, which appear for the first time.

2)          The title in section 2.1 seems too broad. It serves as an introduction to the research area rather than an introduction to the research topic.

3)          The detailed description of the data mentioned in lines 122-126 of P2, "During the summer... the annual total, respectively," should be referenced with the appropriate source.

4)          In lines 245-247 on page 12, relying solely on one year of forecasted numerical data lacks persuasiveness. It is advisable to include multiple statistical indicators or additional forecast years for validation.

5)          I cannot understand the meaning of "six-zoo model" in P13 Line 282.

6)          The formatting of images and text sections still requires careful revision.

 l   Here are some examples in the text

 In line 47 of P1, the capitalization of the initial 'n' in 'non-conventional' is inconsistent. Please check throughout the entire text to ensure consistency in the capitalization of this word.

 In line 156 of P6, "DEMATEL" appears, but it's not the first occurrence, and there's a mix of capitalization. Please review the entire text to maintain consistent naming conventions.

 After line 178 on page 8, the abbreviation "BPNN" is not used. Please ensure consistency in the usage of abbreviations.

 The capitalization of the initial letters in the titles of section 2.4.2 is inconsistent.

 The formatting of the section titles in section 2.5 is not consistent.

 The A matrix mentioned in line 213 on page 10 could be considered for inclusion in the appendix.

 In line 219 of page 10, the description of the image uses "Fig." while the preceding text uses the full term "Figure." Please maintain consistency.

 l   Here are some examples in Figure and Table

The term "reach" in Fig. 2 should be explained and clarified after the image to indicate what it represents.

 The expression in Fig. 3 needs to be streamlined. Additional explanations can be provided after the image rather than stacking text on the image itself.

 The definition of "xi" in Fig. 3 does not need to be repeated.

 The meaning of the letters "ω," "Z," and "α" in Figure 5 appears to be unexplained in the text.

 Please simplify the flowchart in Figure 6 to highlight key points, and ensure consistency in the capitalization of titles between "" and "".

 The numbers following "X" in Figure 7 are not subscripted.

 Consider labeling the image above Figure 8 as it lacks a number. Additionally, due to the presence of multiple subplots, it might be advisable to split it into two separate figures.

 The number "2.6028" in Figure 9 lacks supplementary explanation.

 The latitude and longitude grid is missing in Figure 10.

Comments on the Quality of English Language

Moderate editing of English language required.

Author Response

Dear Expert Teacher.

Greetings. First of all, thank you very much for taking time out of your help to review my thesis. Next, I would like to answer the questions that you have asked me.

  1. In lines 95-98 of P2, the statement "The control group..." mentions various treatment methods without specifying how these methods differ from previous research methods, what problem they aim to address, and it doesn't clarify the full names of abbreviations such as DEMATEL, BP, GA, PSO, which appear for the first time.

Re:In lines 100 to 109 of the article, I wrote the full names of the abbreviations DEMATEL, BP, GA, PSO, etc., which are highlighted in red in the article. In addition to this, I have added the reasons for using them, which are highlighted in green in the article.

2.The title in section 2.1 seems too broad. It serves as an introduction to the research area rather than an introduction to the research topic.

Re:I have changed the title of section 2.1 to " Overview of water resources in Heilongjiang Province".

3.The detailed description of the data mentioned in lines 122-126 of P2, "During the summer... the annual total, respectively," should be referenced with the appropriate source.

Re:In line 148 of the article, below Figure 2, I added the data sources. The data is from the "Heilongjiang Province Water Resources Bulletin", highlighted in green.

4.In lines 245-247 on page 12, relying solely on one year of forecasted numerical data lacks persuasiveness. It is advisable to include multiple statistical indicators or additional forecast years for validation.

Re:In this study, six models are used to predict the unconventional water resources utilisation in Heilongjiang province from 2008 to 2022, and then the actual utilisation of unconventional water resources in 2022 is used as the evaluation criterion, and a set of results with the smallest error from the actual value is selected. And the errors between the actual utilisation of unconventional water resources and the predicted values from 2008 to 2021 are represented in Figure 8.

  1. I cannot understand the meaning of "six-zoo model" in P13 Line 282.

Re:I'm very sorry, the text should read "six sets model", I've made the change and highlighted it in red.

  1. The formatting of images and text sections still requires careful revision.

Re:I made changes to the formatting of the article.

  1. In line 47 of P1, the capitalization of the initial 'n' in 'non-conventional' is inconsistent. Please check throughout the entire text to ensure consistency in the capitalization of this word.

Re:I checked the entire text to make sure the word was case-sensitive.

  1. In line 156 of P6, "DEMATEL" appears, but it's not the first occurrence, and there's a mix of capitalization. Please review the entire text to maintain consistent naming conventions.

Re:In line 170 of the article, I changed "DEMATEL" to "Dematel", which is shown in red in the text.

9.After line 178 on page 8, the abbreviation "BPNN" is not used. Please ensure consistency in the usage of abbreviations.

Re:After line 178, I use the abbreviation "BPNN", as in lines 199, 205 and 206, which are in red in the text.

10.The capitalization of the initial letters in the titles of section 2.4.2 is inconsistent.

Re:Initial case has been changed in the title of section 2.4.2 and is indicated in red in the text.

11.The formatting of the section titles in section 2.5 is not consistent.

Re:I have modified the formatting of the section headings in Section 2.5 to indicate them in red in the text.

12.The A matrix mentioned in line 213 on page 10 could be considered for inclusion in the appendix.

Re:I deleted the A matrix.

  1. In line 219 of page 10, the description of the image uses "Fig." while the preceding text uses the full term "Figure." Please maintain consistency.

Re:I replaced the descriptions of the pictures with "Figure".

14.The term "reach" in Fig. 2 should be explained and clarified after the image to indicate what it represents.

Re:I have explained "Reach", which is shown in red in the text.

15.The expression in Fig. 3 needs to be streamlined. Additional explanations can be provided after the image rather than stacking text on the image itself.

 The definition of "xi" in Fig. 3 does not need to be repeated.

Re:Figure 3 shows the calculation process of Improvement of the Dematel model, so I think the formula in the figure is indispensable. And "X1-X9" in the figure represents nine indicators respectively.

  1. The meaning of the letters "ω," "Z," and "α" in Figure 5 appears to be unexplained in the text.

Re:I have explained the letters in the diagram, which are shown in red in the text.

  1. Please simplify the flowchart in Figure 6 to highlight key points, and ensure consistency in the capitalization of titles between "Ⅰ" and "Ⅱ".

Re:I have added explanations of the images below each image in Figure 6, in green in the text.

18.Consider labeling the image above Figure 8 as it lacks a number. Additionally, due to the presence of multiple subplots, it might be advisable to split it into two separate figures.

Re:I split figure 8 into two figures.

  1. The number "2.6028" in Figure 9 lacks supplementary explanation.

Re:In Figure 10, I have explained the number "2.6028", which is shown in red in the text.

20.The latitude and longitude grid is missing in Figure 10.

Re:I have redrawn Figure 11 with the addition of the latitude and longitude grids.

21.In addition to this, in lines 45-46 of the article, there is a description of the classification of non-conventional water resources, which is shown in red in the article; in lines 46-48 of the article, there is a description of the aspects of the use of non-conventional water resources, which is shown in green in the article; and in the Discussion section of the article, I have added a description of the findings of the results of this study and an outlook of the future direction of the research, which is shown in green in the article.

Reviewer 2 Report (New Reviewer)

Comments and Suggestions for Authors

1. Authors must improve the standard configuration of the text displayed by the Journal (font sizes in titles, text, etc. References are not presented in the journal format).

2. The writing of excerpts from the manuscript (since the Introduction) needs to be improved to avoid explicit repetitions (mainly nouns and adjectives).

3. Some important information must be included in the Introduction:

"non-conventional water" What types?

"methods of Non-conventional water resources" What are the methods?

4. In the phrase "Wu considered water resource endowment conditions and...", indicate Wu's reference;

5. Describe all abbreviations/symbols that appear for the first time in the manuscript (present in the text and figures);

6. How important is section 2.1. Overview of the Research Topic for Materials and Methods? Authors should review this excerpt. This is information about the region under study that apparently could be used in another section. Little use is made of the text as Materials and Methods.

7. Indicate the references for using the equations in Table 1 and Figure 3 for each parameter considered.

8. In Materials and Methods, better describe the method for producing the scoring matrix A.

9. The authors present the results in a descriptive way, with many images. In any case, the discussion could be improved to meet the objective of the work. The discussions are quite general and do not refer much to the particular results found. Authors should substantially improve the presentation of the discussion section.

Author Response

Dear Expert Teacher.

Greetings. First of all, thank you very much for taking time out of your help to review my thesis. Next, I would like to answer the questions that you have asked me.

  1. Authors must improve the standard configuration of the text displayed by the Journal (font sizes in titles, text, etc. References are not presented in the journal format).

 Re: the text of the article I have changed.

  1. The writing of excerpts from the manuscript (since the Introduction) needs to be improved to avoid explicit repetitions (mainly nouns and adjectives).

 Re: the changes I made to the article regarding nouns, etc.

  1. Some important information must be included in the Introduction:

"non-conventional water" What types?

"methods of Non-conventional water resources" What are the methods?

 Re: In lines 45-46 of the article, there is a description of the classification of non-conventional water resources, indicated in red in the article.

In lines 46-48 of the article, there is a description of the use of non-conventional water resources, indicated in green in the article.

  1. In the phrase "Wu considered water resource endowment conditions and...", indicate Wu's reference;

 Re: In lines 66-71 of the article, all references are to "wu" studies, and the references are [18]. The references are in red in the text.

  1. Describe all abbreviations/symbols that appear for the first time in the manuscript (present in the text and figures);

 Re: in lines 100 to 109 of the article, I wrote the full names of the acronyms DEMATEL, BP, GA, PSO, etc., which are highlighted in red in the article. In addition to that, I added the reasons for using them, highlighted in green in the article.

  1. How important is section 2.1. Overview of the Research Topic for Materials and Methods? Authors should review this excerpt. This is information about the region under study that apparently could be used in another section. Little use is made of the text as Materials and Methods.

 Re: I changed the title of section 2.1 to " Overview of water resources in Heilongjiang Province", which belongs to the material part, while sections 2.2, 2.3, 2.4, 2.5 are the research methodology.

  1. Indicate the references for using the equations in Table 1 and Figure 3 for each parameter considered.

 Re: Sorry expert teacher, the formulae in Table 1 are partly from the content of a project in my workplace, which has not been published yet, so there is no reference. I have added the reference to Figure 3, which is on line 180 in the text, with lines 442-444 in green.

  1. In Materials and Methods, better describe the method for producing the scoring matrix A.

 Re: On the advice of another reviewer, in line 232 of the article, I deleted matrix A. The approach to the generation of matrix A is specifically described in Figure 3.

  1. The authors present the results in a descriptive way, with many images. In any case, the discussion could be improved to meet the objective of the work. The discussions are quite general and do not refer much to the particular results found. Authors should substantially improve the presentation of the discussion section.

Re: In the Discussion section of the article, I have added a discussion of the findings of the results of this study and an outlook on future research directions, which is indicated in green in the article.

10.In addition, in line 148 of the article, below Figure 2, I added the data sources. The data is from "Heilongjiang Province Water Resources Bulletin", which is highlighted in green; in line 170 of the article, I changed "DEMATEL" to "Dematel", which is highlighted in red; after line 178, I used the abbreviation "BPNN". ", highlighted in red in the text; after line 178, I used the abbreviation "BPNN", as in lines 199, 205 and 206, highlighted in red in the text; in the title of section 2.4.2, I changed the case of the first letter, highlighted in red in the text; I modified the first letter of "Reach" in Fig. 2, and the second letter of "Reach" in Fig. 2, highlighted in green in the text; I modified the first letter of "Reach" in Fig. 2, highlighted in red in the text. "Reach" in Figure 2, in red in the text; I explained the letters in Figure 5, in red in the text; I added explanations of the pictures below each picture in Figure 6, in green in the text; in Figure 10, I explained the number "2.6028 ", which is explained in red in the text.

Round 2

Reviewer 1 Report (Previous Reviewer 1)

Comments and Suggestions for Authors

The authors focus on studying the influences factor on Heilongjiang Province's unconventional Water Resources. This study introduces the concept of comparative testing and employs enhanced Dematel and random forest models to identify two optimal indicator sets from a pool of nine indicators. Based on these two best indicator sets, three prediction models - BP neural network, PSO-BP, and GA-BP - were utilized to forecast the future potential of unconventional water resource utilization in Heilongjiang Province. I recommend acceptance for the paper. 

Author Response

Dear Expert Teacher.

Greetings. Firstly, thank you very much for taking time out of your busy schedule to review my thesis. Secondly, I have improved the formulas in Table 1 by explaining the meaning of each formula and adding references, which are shown in red in the text.

作者重点研究了黑龙江省非常规水资源的影响因素。本研究引入了比较测试的概念,并采用增强的Dematel和随机森林模型,从九个指标池中确定两个最佳指标集。基于这两个最佳指标集,利用BP神经网络、PSO-BP和GA-BP3个预测模型预测了黑龙江省未来非常规水资源利用潜力。我建议接受这篇论文。

回应:非常感谢您对本论文的支持,也非常感谢您对本论文的意见和建议,这将大大有助于提高我论文的水质,再次衷心感谢您,祝您身体健康,工作顺利。

Reviewer 2 Report (New Reviewer)

Comments and Suggestions for Authors

in table 1, the authors inform that the formulas, extremely important for the results presented, were developed by them and do not present references. On the other hand, they only present the meaning of each parameter and do not highlight/explain the nature of each function. Why use those specific types of functions and not others? How can results be presented based on the result of applications of formulas that are not explained in detail and have no references? It is somewhat confusing to accept a theory without explaining why it is used.

For the work to have a scientific sound, these details must be described. I suggest that in the Table 1 section, each unprecedented formula must be explained to readers (why is each of them there?) of the area for the purposes of validity and acceptance of the results obtained .

The use of formula references of similar nature for similar purposes is highly required.

Author Response

Dear Expert Teacher.

Greetings. First of all, thank you very much for taking time out of your help to review my thesis. Next, I would like to answer the questions that you have asked me.

in table 1, the authors inform that the formulas, extremely important for the results presented, were developed by them and do not present references. On the other hand, they only present the meaning of each parameter and do not highlight/explain the nature of each function. Why use those specific types of functions and not others? How can results be presented based on the result of applications of formulas that are not explained in detail and have no references? It is somewhat confusing to accept a theory without explaining why it is used.

For the work to have a scientific sound, these details must be described. I suggest that in the Table 1 section, each unprecedented formula must be explained to readers (why is each of them there?) of the area for the purposes of validity and acceptance of the results obtained .

The use of formula references of similar nature for similar purposes is highly required.

RESPONSE: Hello Dear Expert Teacher, Thank you very much for your advice again on this paper, which has helped me significantly to improve this paper. Regarding your suggestion that there are no references for the formulas in Table 1, I have improved this part. I have explained the nature of each formula and attached references. They are highlighted in red in the text. Again, thank you very much for your suggestion.

Round 3

Reviewer 2 Report (New Reviewer)

Comments and Suggestions for Authors

.

This manuscript is a resubmission of an earlier submission. The following is a list of the peer review reports and author responses from that submission.

Round 1

Reviewer 1 Report

Comments and Suggestions for Authors

The authors focus on studying the influences factor on Heilongjiang Province's unconventional Water Resources. This work introduces the concept of comparison testing and utilizes enhanced Dematel and random forest models to screen two sets of optimal indicators out of nine indicators. Based on the two best indicator sets, three prediction models - BP neural network, PSO - BP, and GA - BP - were used to forecast the future potential of unconventional water resource use in Heilongjiang Province. It is essential to study. However, too many concerns should be addressed. We decided to reject this paper.

Here are our major comments:

1)       The writing logic and structure of the article are very confusing. Authors should logically organize each work and result instead of describing all the relevant work.

2The English must be polished by a native speaker. Most of the sentences are not connected.

3The author seems not to have paid enough attention to this paper. Some mistakes in the reference format. The font size in some figures is extremely unreadable.

Here are some examples in the text

abbreviations

         P1, Line 17  BP neural network, PSO - BP, and GA - BP -

         P1, Line 36 . The Global Environment Outlook (GEO) is not needed for abbreviations

Introduction

Lines 57-62 The key novelty of this work is missing. The first paragraph does not show it clearly due to unnecessary details.

Lines 108-112 The research background is unclear as well.

Lines 114-126 At the end of your introduction, an evident outline or target should be placed to show the logic of this work.

Materials and Methods

Line 129-130 Materials and Methods

Same issue: too many unnecessary details in the study region and methods. A simple introduction should be done with a clear structure. This work is a research for investigation rather than proposing a novel method. The method is not the key one.

Suggestion: a map can help you introduce the study region.

Lines 165 (김준순 et al.,2003).

Results and Conclusion

This work seems to be a data report rather than an investigation.

Line 278 Matlab->MATLAB

Line 291 the training set-up can be placed in the supplement

Line 300-301 the way you plot does not matter

l   Here are some examples in Figure

Figure 2

vertical text in Figures 3, 5, and 8 is hard to

the red circle marker in Figure 6

Comments on the Quality of English Language

The English must be polished by a native speaker. Most of the sentences are not connected.

Author Response

尊敬的专家。

问候。非常感谢您为本文提供建议的能力,我结合两位专家提出的建议进行了以下修改。

1.In 文章第2行、第3行,我将本文标题改为“中国黑龙江省非常规水资源现状与未来利用分析”(修订文件中红色突出显示)

2.In 引言的第 17 行和第 18 行,我消除了 PSO-BP 和 GA-BP 的使用。(在修订文件中以绿色突出显示)

3.我将“非常规水”替换为“非常规水”(在修改文件中以黄色突出显示)

4.In 文章的第35行,我删除了《全球环境展望》的缩写“GEO”。

5.In 文章的第 43 行和第 44 行,我将非常规水资源的分类更改为“处理过的废水、处理过的农业排水、淡化水、苦咸地下水、降雨收集”。(在修订文件中以蓝色突出显示)。收获雨水......”(在修订文件中以蓝色突出显示)

6.In 文章的第 58 行到第 78 行,我添加了有关黑龙江省水平衡的信息,解释了为什么需要非常规用水。(修订文件中以红色突出显示)

7.In 文章第99-101行,我补充道:“最近的研究表明,经济因素是影响非常规水资源利用的关键因素。(在修订文件中以红色突出显示)

8.In 文章的第 127 行,我重新绘制了图 1 并停止使用缩写。

9.在文章的第160行,我添加了一张黑龙江省水系的地图。

10.In 文章的第 279 行到第 284 行,我添加了对图形的解释。(在修订文件中以红色突出显示)

11.In 文章的第 140 行到第 159 行,我删除了研究区域部分中不必要的内容。(修订文件中以绿色突出显示)

12.In 文章第2.3节和第2.4节中,增加了对关键指标进行筛选和非常规用水潜力预测研究的原因,以增强文章的逻辑性,并使各节之间更加连贯。(在修订后的文件中以红色突出显示)

13.In 文章的材料和方法部分,我重写了研究方法并删除了冗余。(在修订后的文件中以黄色突出显示)

14.In 本文的第 3.2 节中,我删除了冗余。(修改文件中以红色突出显示)

15.In 文章的第4节中,我补充了黑龙江省非常规水资源利用的重要性。(修订文件中以红色突出显示)

16.In 文章的引言部分(第 129 行到第 137 行),我添加了文章研究的目的。(在修订文件中以红色突出显示)

17.In 参考文献中,我删除了这些数字

18.In 文章的介绍部分(第 79 行到第 93 行),我重写了研究背景。(在修订后的文件中以红色突出显示)

19.除此之外,我还改进了英语。

Reviewer 2 Report

Comments and Suggestions for Authors

1)         I recommend changing the title to: (The Analysis of Present and Future Use of Non-conventional Water Resources in Heilongjiang Province, China).

2)         Do not use the abbreviations in the abstract such as PSO – BP, and GA – BP, etc. ….

3)         The English language should be revied by native English speaker.

4)         Replace the word (Unconventional) with (non-conventional) across the manuscript.  

5)         The references should be cited in the text as [number] for example [1], [2], …etc.

6)         On page (2) line (46) the classifications of the non-conventional water resources should be as follows:

·          Treated wastewater.

·          Treated agricultural drainage.

·          Desalinated water.

·          Brackish groundwater.

·          Harvesting of rainfall

7)         On page (2) add some information about the water balance for Heilongjiang Province and why there is a need for non-conventional water resources use?

8)         On page (3), line (101) it should be: Recent studies indicating that the economic ……………… Then add more than one reference.

9)         On page (2), Figure (1) do not use abbreviations (X1, X2, ….), use the real names Economic Benefit, economic costs, …… etc.

10)     On page (4) line (135) add the general location map for the Heilongjiang Province and the existing rivers.

11)     The same in all figures use the real names.

12)     Figure (9) on page (13) it is not clear and need more explanation.

13)     In all references remove the numbers [1], [2], …….

14)     I recommend removing all old references to minimize the references list to less than two pages.

15)     In the recommendation add some economic and environmental justifications for the importance of using non-conventional water resources in Heilongjiang Province.   

Comments on the Quality of English Language

The English language should be revied by native English speaker.

Author Response

Dear Experts.

Greetings. Thank you very much for your ability to advise on this paper, and I have made the following changes in conjunction with the suggested changes made by the two experts.

1.In lines 2 and 3 of the article, I replaced the title of this article with“The Analysis of Present and Future Use of Non-conventional Water Resources in Heilongjiang Province, China”(Highlighted in red in the revision document)

2.In lines 17 and 18 of the introduction, I eliminated the use of PSO-BP and GA-BP . (Highlighted in green in the revision file)

3.I replaced "unconventional water" with "non-conventional water" (highlighted in yellow in the modification file)

4.In line 35 of the article, I deleted the abbreviation "GEO" for Global Environment Outlook.

5.In lines 43 and 44 of the article, I changed the categorization of non-conventional water resources to "Treated wastewater, Treated agricultural drainage, Desalinated water, Brackish groundwater, Harvesting of rainfall." (highlighted in blue in the revision file). Harvesting of rainfall..." (highlighted in blue in the revision document)

6.In lines 58 through 78 of the article, I have added information about the water balance in Heilongjiang Province that explains why unconventional water use is needed. (Highlighted in red in the revision document)

7.In lines 99-101 of the article, I added "Recent studies indicating that economic factors are key factors influencing the utilization of non-conventional water resources." (highlighted in red in the revision document)

8.In line 127 of the article, I redrew Figure 1 and stopped using abbreviations.

9.On line 160 of the article, I added a map of the water system in Heilongjiang.

10.In lines 279 through 284 of the article, I added an explanation of the graphic. (Highlighted in red in the revision file)

11.In lines 140 through 159 of the article, I removed unnecessary content in the study area section. (Highlighted in green in the revision document)

12.In sections 2.3 and 2.4 of the article, the reasons for conducting the screening of key indicators and the forecasting study of the potential for non-conventional water use have been added in order to enhance the logic of the article and to make the sections more coherent with each other. (Highlighted in red in the revised document)

13.In the Materials and Methods section of the article, I rewrote the research methodology and removed redundancies. (Highlighted in yellow in the revised document)

14.In section 3.2 of the article, I removed the redundancy. (Highlighted in red in the modification document)

15.In section 4 of the article, I added the significance of the use of non-conventional water resources in Heilongjiang Province. (Highlighted in red in the revision document)

16.In 文章的引言部分(第 129 行到第 137 行),我添加了文章研究的目的。(在修订文件中以红色突出显示)

17.In 参考文献中,我删除了这些数字

18.In 文章的介绍部分(第 79 行到第 93 行),我重写了研究背景。(在修订后的文件中以红色突出显示)

19.除此之外,我还改进了英语。

Round 2

Reviewer 1 Report

Comments and Suggestions for Authors

The authors focus on studying the influences factor on Heilongjiang Province's unconventional Water Resources. This study introduces the concept of comparative testing and employs enhanced Dematel and random forest models to identify two optimal indicator sets from a pool of nine indicators. Based on these two best indicator sets, three prediction models - BP neural network, PSO-BP, and GA-BP - were utilized to forecast the future potential of unconventional water resource utilization in Heilongjiang Province. However, many concerns proposed in our first review have been ignored, which should be addressed. Please check our comments carefully and reply my comments formally in a format text (.pdf or .docx). Therefore, we decided to reject this paper.

Comments on the Quality of English Language

The English must be polished by a native speaker. Most of the sentences are not connected. 

Reviewer 2 Report

Comments and Suggestions for Authors

It has been modified and improved according to the requirements and basically reached a level that can be published in the journal.

Comments on the Quality of English Language

Please carefully review the professional vocabulary involved in the full text before publication.